# High frequency of germline recombination in Nestin-Cre transgenic mice crossed with Glucagon-like peptide 1 receptor floxed mice

**Yusuke Kajitani, Takashi Miyazawa** ⊙ *, **Tomoaki Inoue, Nao Kajitani, Masamichi Fujita, Yukina Takeichi, Yasutaka Miyachi** ⊙ **, Ryuichi Sakamoto, Yoshihiro Ogawa**

Department of Medicine and Bioregulatory Science, Graduate School of Medical Sciences, Kyushu University, Fukuoka, Japan

* miyazawa.takashi.975@m.kyushu-u.ac.jp

**Data Availability Statement:** All relevant data are within the paper and its Supporting information files.

## Abstract

The Cre-loxP strategy for tissue-specific gene inactivation has become a widely employed tool in several research studies. Conversely, inadequate breeding and genotyping without considering the potential for non-specific Cre-recombinase expression may lead to misinterpretations of results. Nestin-Cre transgenic mice, widely used for the selective deletion of genes in neurons, have been observed to have an incidence of Cre-line germline recombination. In this study, we attempted to generate neuron-specific Glucagon-like peptide 1 receptor (Glp1r) knock-out mice by crossing mice harboring the Nestin-Cre transgene with mice harboring the Glp1r gene modified with loxP insertion, in order to elucidate the role of Glp1r signaling in the nervous system. Surprisingly, during this breeding process, we discovered that the null allele emerged in the offspring irrespective of the presence or absence of the Nestin-Cre transgene, with a high probability of occurrence (93.6%). To elucidate the cause of this null allele, we conducted breeding experiments between mice carrying the heterozygous Glp1r null allele but lacking the Nestin-Cre transgene. We confirmed that the null allele was inherited by the offspring independently of the Nestin-Cre transgene. Furthermore, we assessed the gene expression, protein expression, and phenotype of mice carrying the homozygous Glp1r null allele generated from the aforementioned breeding, thereby confirming that the null allele indeed caused a global knock-out of Glp1r. These findings suggest that the null allele in the NestinCre-Glp1r floxed breeding arose due to germline recombination. Moreover, we demonstrated the possibility that germline recombination may occur not only during the spermatogenesis at testis but also during epididymal sperm maturation. The striking frequency of germline recombination in the Nestin-Cre driver underscores the necessity for caution when implementing precise breeding strategies and employing suitable genotyping methods.

## Introduction

The Cre-loxP recombination technique is frequently employed to investigate the cell-specific effects of gene expression. This methodology capitalizes on the functional properties of the bacteriophage P1 topoisomerase, Cre-recombinase, which specifically recognizes 34 base pair

**Funding:** TM received the Japan Society for the Promotion of Science KAKENHI Grant Number 22K08673. TM also received Smoking Research Foundation. The funders had no role in study design, data collection and analysis, decision to publish, or preparation of the manuscript.

**Competing interests:** The authors have declared that no competing interests exist.

loxP sites [1]. Cre-recombinase facilitates the deletion of DNA fragments that are flanked by a pair of loxP sites or flips DNA fragments between two inverted loxP sites. Breeding floxed mice (carrying the gene of interest flanked by loxP sites) with Cre mice (expressing Cre-recombinase driven by a cell-specific promoter) allows the generation of conditional knock-out or knock-in mice [2, 3]. Since its inception, a plethora of Cre drivers have been developed, and a diverse range of genetically targeted animal models have been utilized in biomedical research. Despite its widespread usage, this technique has faced criticism due to several reported issues and limitations, including Cre toxicity and unexpected recombination resulting from non-specific Cre activation [4–8].

Nestin is an intermediate neurofilament protein that was initially reported as a marker of neuroepithelial stem cells. Thus, Nestin-Cre transgenic mice are widely used for investigating the nervous system [9, 10]. Glucagon-like peptide-1 (Glp1) is secreted from intestinal L-cells and the brain and functions through hormonal and neural pathways to regulate islet function, appetite, and gut motility [11, 12]. We attempted to generate neuron-specific Glp1r knock-out mice by breeding Nestin-Cre mice with Glp1r floxed mice to clarify the role of Glp1-Glp1r signaling in the nervous system. However, we observed unexpected genotyping results. Several reports have described similar phenomena in the Nestin-Cre line and revealed that it could be due to germline recombination [13, 14]. However, the frequency of this phenomenon varied widely among reports, and the precise cause remains elusive.

In this study, we demonstrated that unexpected genotyping results observed when crossing Nestin-Cre mice with Glp1r floxed mice were caused by germline recombination, which occurred with a remarkably high frequency. Additionally, we identified the stages during gametogenesis where this recombination event takes place and elucidated the underlying mechanisms. These findings highlight the importance of being vigilant during breeding and genotyping, and the need to identify and address issues when using the Cre-loxP technique.

## Results

### Genotyping methods and identification of offspring genotypes

Genotyping was performed through PCR analysis using genomic DNA (gDNA) extracted from tail biopsies. Two genotyping methods were employed to discriminate among Glp1r alleles: one for distinguishing the wild-type (wt) allele from the floxed allele and another for identifying the knock-out allele. To detect the wt allele and floxed allele, a primer set was designed to amplify the region flanking the 3' loxP site, enabling the detection of the wt and floxed alleles (primer set 1) (Fig 1A). Specifically, the wt allele yielded a 273 bp product, while the floxed allele produced a 416 bp product (Fig 1B). To detect the knock-out allele, another primer set was employed, targeting the entire floxed region (primer set 2) (Fig 1A). With primer set 2, the knockout allele was identified by a 423 bp product, while the wt allele and the floxed allele were identified in the vicinity of a 2000 bp product (Fig 1B). For the Nestin-Cre allele, we used the primer sets distinguishing the Cre positive allele (150 bp) and the Cre negative allele (246 bp). The identification of offspring genotypes was conducted by these three genotyping analyses (Fig 1B). To generate neuron-specific Glp1r knockout mice (NesCre-Glp1r fl/fl), we initially crossed NesCre-Glp1r wt/wt mice with Glp1r fl/fl mice, resulting in the generation of NesCre-Glp1r fl/wt mice (F1). Subsequently, we attempted to generate NesCre-Glp1r fl/fl mice by crossing NesCre-Glp1r fl/wt mice with Glp1r fl/fl mice (F2). According to Mendelian genetics, this mating scheme theoretically should have resulted in four distinct genotypes: NesCre-Glp1r fl/fl, NesCre-Glp1r fl/wt, Glp1r fl/fl, and Glp1r fl/wt mice. However, in our actual breeding experiments, in addition to these four genotypes, NesCre-Glp1r fl/- and Glp1r fl/- mice were also born (Fig 1C).

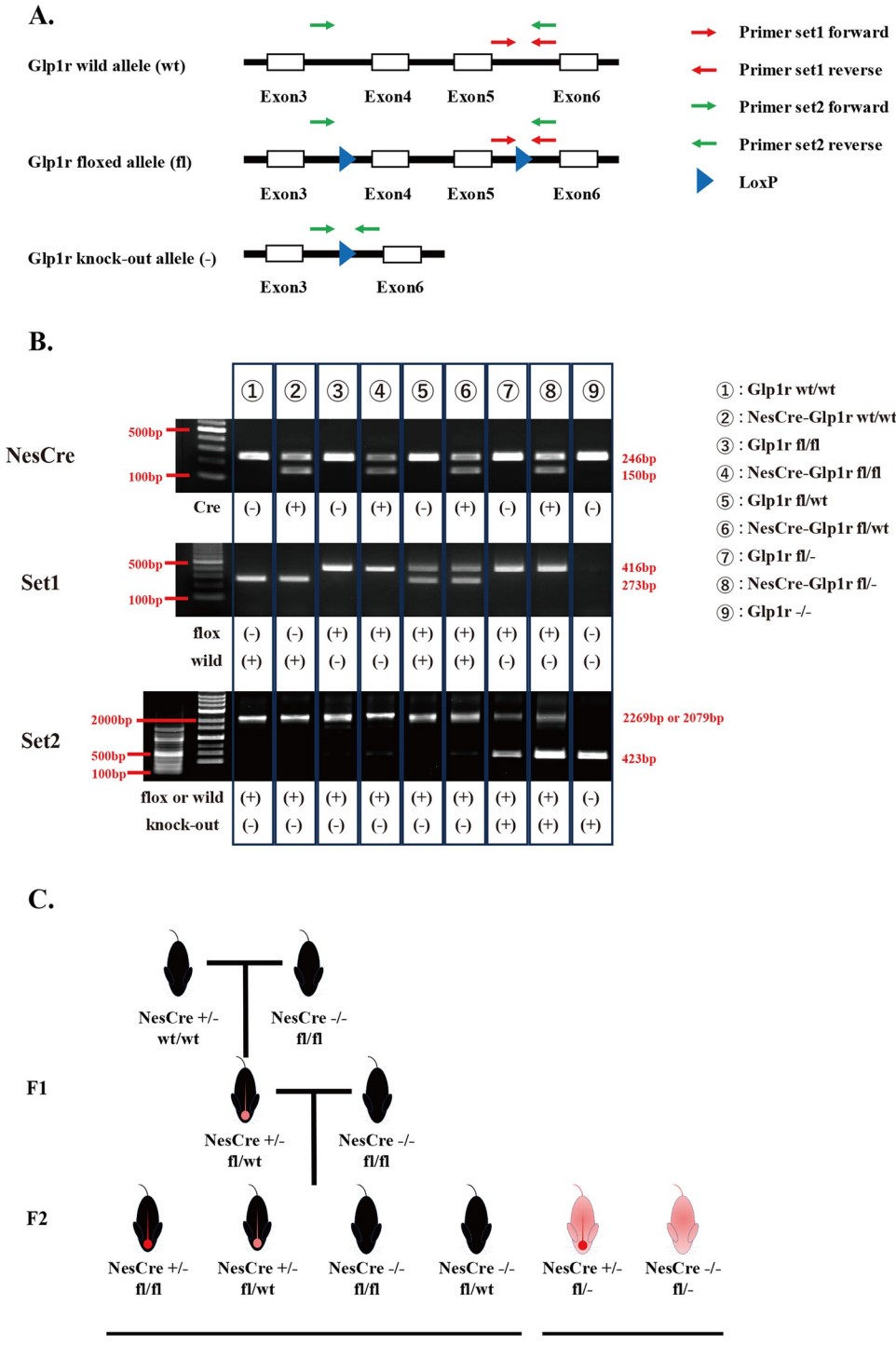

**Fig 1. Genotyping strategy using tail samples for Glp1r wild-type allele, floxed allele, knock-out allele, and Nestin-Cre allele.** (A) Top line is the wild-type (wt) allele. Middle line is the floxed allele with the exon flanked by loxP sites (arrowheads). Underline is the floxed allele after Cre-mediated recombination. The location of PCR primer set1 and set2 are indicated. (B) The combination of genotyping for each genotype. PCR genotyping products using Nestin-Cre primer: wt allele is shown as 246 bp, Cre allele as 150 bp. PCR genotyping products using primer set1: wt allele presented as 273 bp, and floxed allele as 416 bp. PCR genotyping products using primer set2: wt allele is shown as 2079 bp, floxed allele as 2269 bp, and knock-out allele as 423 bp. We use a 100 bp marker for the identification of product size. Regarding genotyping with primer set2, we also simultaneously use a 1000 bp marker (right side) in addition to a

100 bp marker (left side). NesCre: Nestin-Cre allele, fl: floxed allele, wt: wild-type allele, -: knock-out allele. Underlying raw images can be found at S1 Raw images. (C) Schematic of breeding strategies using Nestin-Cre and Glp1r flox transgenic mice to generate male and female breeders with a floxed Glp1r in Nestin-promoter driven tissue and observed progeny included expected and unexpected genotypes from NesCre-Glp1r fl/wt mice crossed with Glp1r fl/fl mice.

## Parental Cre-mediated effects influence knock-out allele generation in NestinCre-Glp1r floxed mice breeding

In mice classified as the Glp1r fl/- genotype at F2, which lacks the Cre allele, it is conceivable that the knock-out allele in these mice was influenced by parental Cre during gametogenesis or early embryogenesis. To investigate this matter, mice with the Glp1r fl/- genotype were crossed to determine if the knock-out allele could be inherited in the offspring. When Glp1r fl/- mice were bred together, the resulting offspring mice were classified into three genotypes (Glp1r fl/fl, Glp1r fl/-, Glp1r -/-) through genotyping, and their probabilities of occurrence approximately followed the principles of Mendelian genetics, with Glp1r fl/fl at 26.2%, Glp1r fl/- at 51.3%, and Glp1r -/- at 22.5% (Fig 2A). To ascertain whether the presence of knock-out alleles affects embryonic lethality and developmental abnormalities, we compared the differences in litter size, duration from mating to birth, and pre-weaning mortality between Glp1r fl/fl mice mating and Glp1r fl/- mice mating. There were no significant differences in these parameters between genotypes (S1 Table).

## Phenotypic characterization of Glp1r -/- mice: Gene expression, protein levels, and functional assessments

To confirm the phenotype of Glp1r -/- mice generated in this study, we examined the mRNA and protein expression levels of Glp1r. First, we performed quantitative PCR (qPCR) to compare *Glp1r* mRNA expression levels between Glp1r fl/fl and Glp1r -/- mice in various tissues known to express Glp1r [15]. Consistent with our hypothesis, *Glp1r* mRNA expression was undetectable in various tissues including the brain, colon, kidney, heart, lung, pancreas, and stomach in mice with a Glp1r -/- genotype (Fig 2B). Importantly, among the examined tissues, both the lung and stomach exhibited notably high *Glp1r* gene expression, which aligned with previous findings [16]. Next, we confirmed the expression levels of Glp1r protein in the lung and stomach of Glp1r -/- mice. By western blotting analysis, we observed significantly reduced levels of Glp1r protein in the lung and stomach of Glp1r -/- mice in comparison to control mice (Fig 2C).

In this study, we conducted evaluations of not only gene and protein expression but also the functional aspects of Glp1r -/- mice to demonstrate these mice have the same phenotype as global Glp1r knockout mice described in previous reports. On a standard diet, there was no significant difference in body weight between Glp1r fl/fl and Glp1r -/- mice (Fig 3A). Food intake at 11 weeks also showed no difference between genotypes (Fig 3B). The Glp1r agonist, liraglutide, typically suppresses feeding upon injection. We observed a significant decrease in food intake in Glp1r fl/fl mice after liraglutide administration (Vehicle: -7.7%, Liraglutide: -57.6%), while food intake in Glp1r -/- mice was unaffected (Vehicle: -2.2%, Liraglutide: -1.6%) (Fig 3C). Intraperitoneal glucose tolerance tests (ipGTT) were performed to determine the effect of the Glp1-Glp1r axis on glucose control. Glucose levels at 0 minute did not differ significantly between Glp1r -/- and Glp1r fl/fl mice, but Glp1r -/- mice exhibited higher glucose levels at 15 minutes after glucose injection (Fig 3D). As Glp1 is one of the most potent stimulators of insulin gene expression and secretion, plasma insulin levels were measured

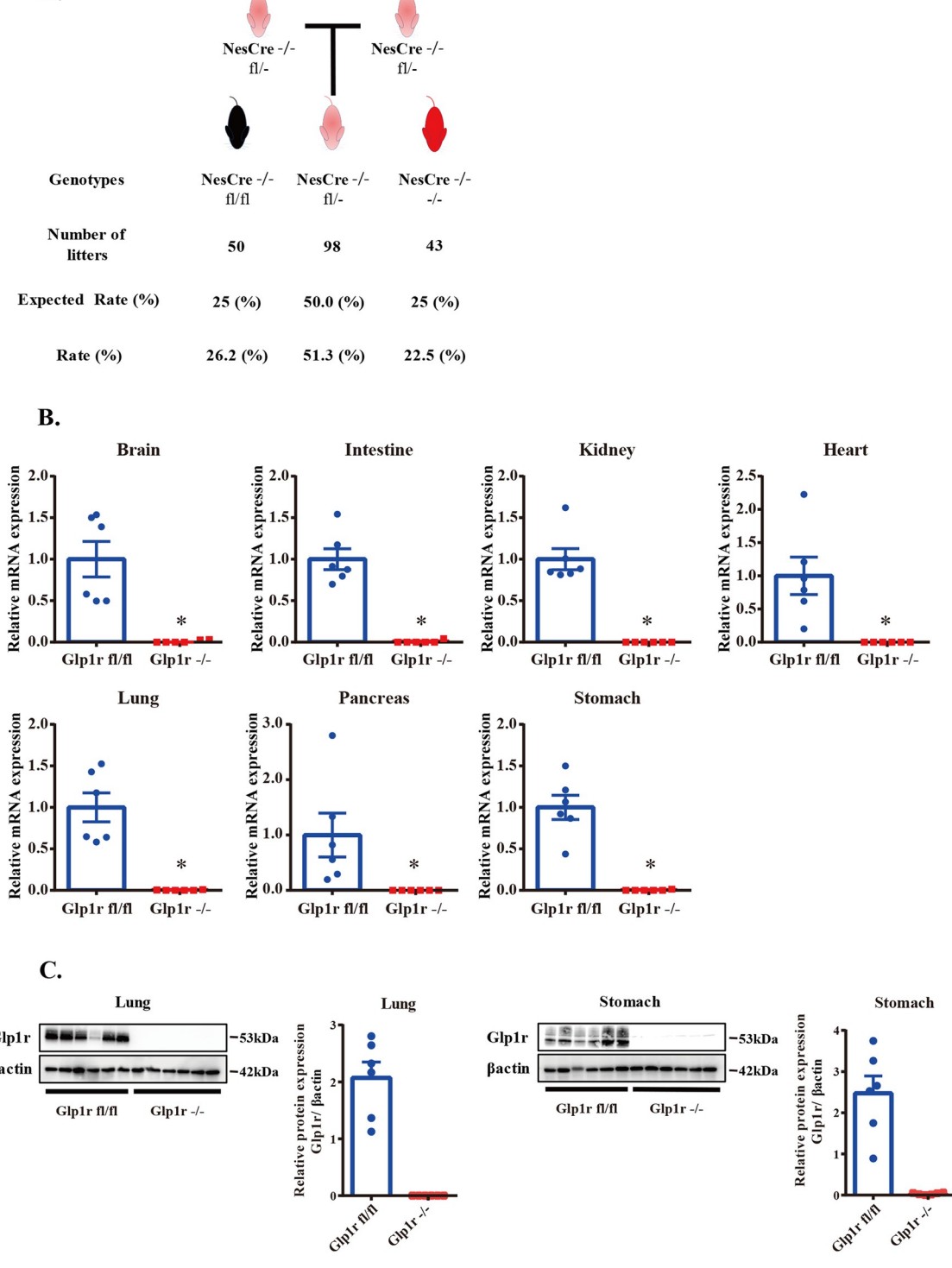

**Fig 2. There was no Glp1r expression in the mice with homozygous knock-out allele.** (A) Expected and observed progeny from hemizygous Glp1r knock-out mice (Glp1r fl/-) crossed with each other. The number and rate of progeny per genotype were illustrated. (B) Quantification of *Glp1r* mRNA level against Glp1r fl/fl mice in the brain, kidney, stomach, pancreas, lung, intestine, and heart. (C) Representative Western blots for the assessment of Glp1r expression and quantitative bar graphs. The expression level of Glp1r is normalized to β-actin. Data represent mean ± SEM, n = 6 mice per group. *$p < 0.05$, statistical analysis was Student's t-test. Individual data can be found at S1 Data and underlying raw images at S1 Raw images.

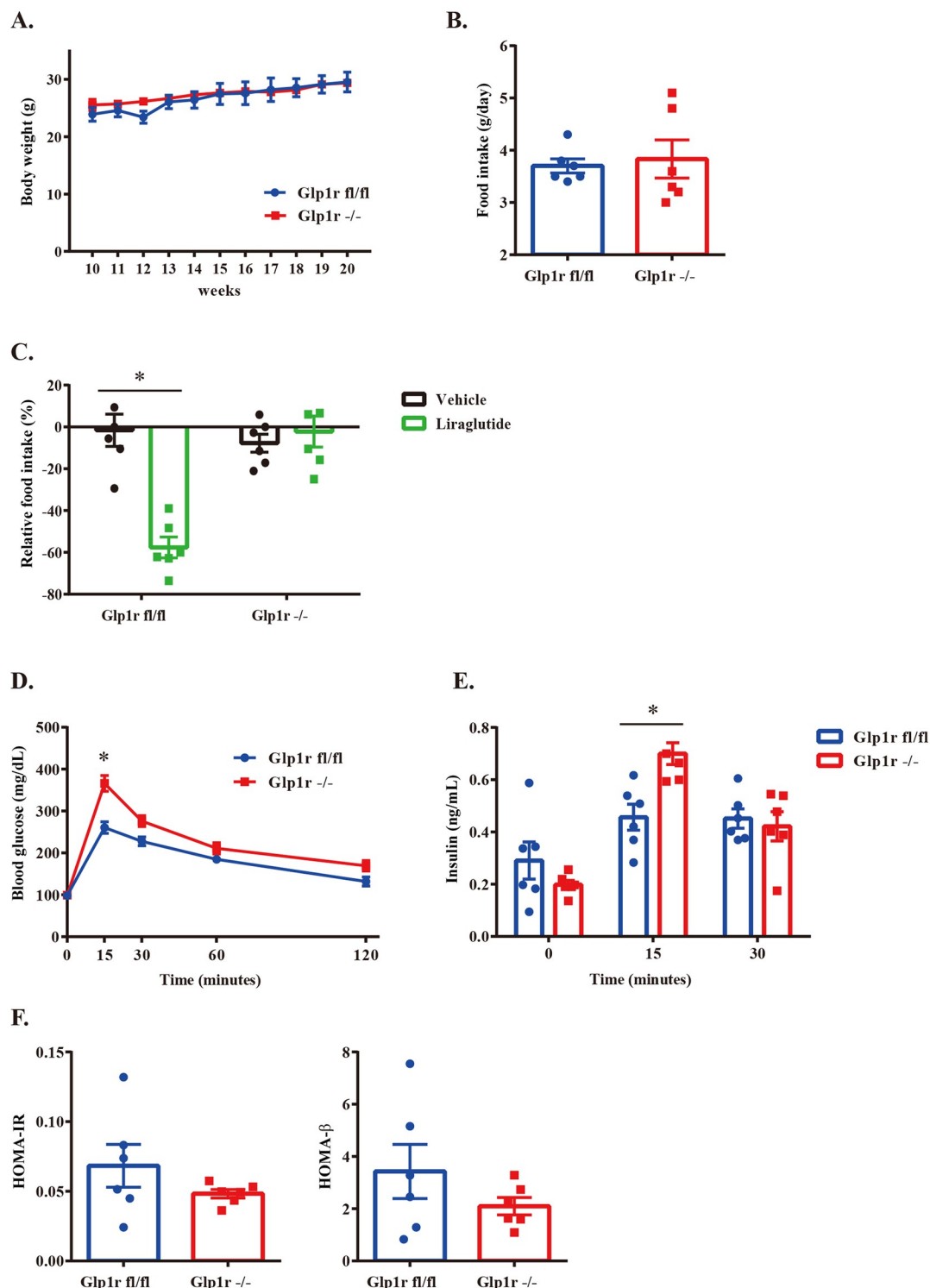

**Fig 3. The feature of Glp1r -/- mice.** (A) Body weight analysis on a standard diet. (B) The average daily food intake at 11 weeks. (C) Anorectic effects of liraglutide (ip injection: 1mg/kg) at 24 hours. (D, E) Blood glucose (D) and plasma insulin (E) levels during ipGTT (Blood glucose: 0, 15, 30, 60, 120 minutes. Plasma insulin: 0, 15, 30 minutes). (F) HOMA-IR and HOMA-β were calculated from the baseline value of ipGTT. Data represent mean ± SEM, n = 6 mice per group. *$p < 0.05$, statistical analysis was two-way repeated measures ANOVA with Bonferroni corrected Student's t-test post-hoc (A)(D)(E); Student's t-test (B)(F); one-way ANOVA with Tukey's HSD test post-hoc (C). Individual data can be found at S1 Data.

during at 0, 15 and 30 minutes after glucose injection. Glp1r -/- mice exhibited higher insulin levels at 15 minutes after glucose injection compared to Glp1r fl/fl mice (Fig 3E). To evaluate glucose tolerance, we calculated homeostasis model assessment estimated insulin resistance (HOMA-IR) calculations, a measure of insulin resistance, and homeostasis model assessment of β-cell function (HOMA-β) calculations, a measure of basal insulin secretion. There was no difference between genotypes in both HOMA-IR and HOMA-β (Fig 3F).

## Frequency and gender differences of germline recombination in NestinCre-Glp1r floxed mice breeding

There have been several studies reporting the frequency of non-specific recombination when using Cre-loxP recombination technic. However, there is considerable variability in the reported frequencies, and some studies have also reported gender differences [13]. To further investigate this phenomenon, we analyzed the number of offspring at the F2 generation with the knock-out allele and the floxed allele. Interestingly, we observed that when one parent carried the Nestin-Cre allele, the frequency of offspring with the knock-out allele was 45.9%, whereas the frequency of offspring with fl/fl genotype was only 3.1%. Consequently, the probability of the floxed allele undergoing germline recombination and being inherited as a knock-out allele in the offspring was 93.6% (Fig 4A). Moreover, when the parent carrying the Cre allele was male, the probability of germline recombination was 86.1%, whereas it was 100% when the Cre-positive parent was female (Fig 4B and 4C). The high frequency of recombination suggests a possibility of underestimating the number of NesCre-Glp1r fl/fl offspring due to embryonic lethality. However, no significant differences were found in terms of litter size, number of days from mating to birth, and the number of mice that died before weaning among different mating patterns examined in our study (S1 Table).

## Detection of somatic recombination in the tail

Nestin, which was generally considered specific to nerves, was reported to have mRNA expression detected in tissues other than neurons as well [17]. Additionally, it is possible that the tail samples used for genotyping may have contained some neural tissue contamination. Therefore, we investigated the potential occurrence of somatic recombination in the tail due to unexpected Cre activity. We compared the expression levels of knock-out alleles in gDNA extracted from the tail of Glp1r fl/-, NesCre-Glp1r fl/wt, and Glp1r fl/wt mice using primer set 2 (Fig 1A). These primers exclusively amplified knock-out alleles since the PCR products from the floxed and wt alleles were too long to be amplified by this qPCR protocol. Additionally, because the primers targeted introns, only gDNA, not mRNA, could be amplified. While Glp1r fl/wt mice did not exhibit any presence of knock-out alleles, NesCre-Glp1r fl/wt mice showed a very low level of knock-out allele expression, approximately 1.5%, compared to Glp1r fl/- mice (Fig 5A). Furthermore, we examined the expression levels of *Nestin* mRNA in the tail using C57BL/6JJcl mice and confirmed minimal expression, approximately 3.6%, compared to the brain (Fig 5B). Next, to evaluate whether the genotyping results obtained from tail samples were influenced by somatic recombination, we conducted genotyping on blood samples. The genotyping results from both tail and blood samples were consistent (Fig 1B and S1 Fig). However, when we examined the expression levels of the knock-out allele between tail and blood samples in NesCre-Glp1r fl/fl and NesCre-Glp1r fl/wt mice, we observed a slight expression of the knock-out allele in tail samples compared to blood samples (Fig 5C). These findings suggest that genotyping in tail samples from mice with both Nestin-Cre and the floxed allele may detect somatic recombination, albeit very slightly.

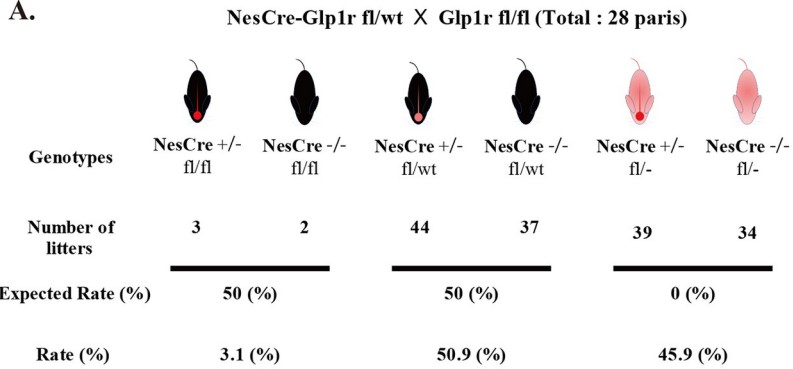

**A.**  **NesCre-Glp1r fl/wt X Glp1r fl/fl (Total : 28 paris)**

| Genotypes | NesCre +/- fl/fl | NesCre -/- fl/fl | NesCre +/- fl/wt | NesCre -/- fl/wt | NesCre +/- fl/- | NesCre -/- fl/- |
|---|---|---|---|---|---|---|
| Number of litters | 3 | 2 | 44 | 37 | 39 | 34 |
| Expected Rate (%) | 50 (%) | | 50 (%) | | 0 (%) | |
| Rate (%) | 3.1 (%) | | 50.9 (%) | | 45.9 (%) | |

**Rate of germline recombination : 93.6 %**

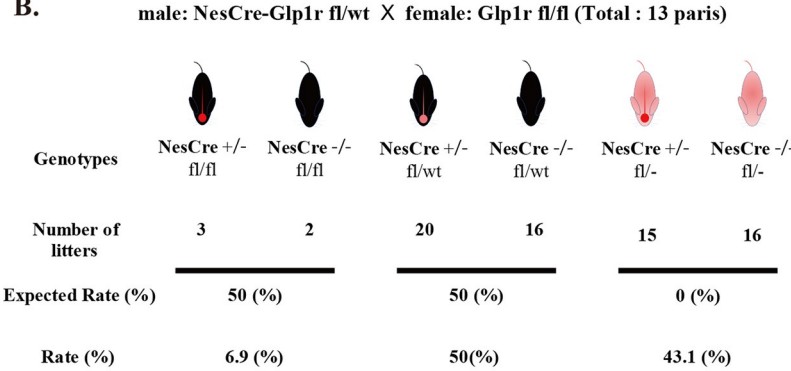

**B.**  **male: NesCre-Glp1r fl/wt X female: Glp1r fl/fl (Total : 13 paris)**

| Genotypes | NesCre +/- fl/fl | NesCre -/- fl/fl | NesCre +/- fl/wt | NesCre -/- fl/wt | NesCre +/- fl/- | NesCre -/- fl/- |
|---|---|---|---|---|---|---|
| Number of litters | 3 | 2 | 20 | 16 | 15 | 16 |
| Expected Rate (%) | 50 (%) | | 50 (%) | | 0 (%) | |
| Rate (%) | 6.9 (%) | | 50(%) | | 43.1 (%) | |

**Rate of germline recombination : 86.1 %**

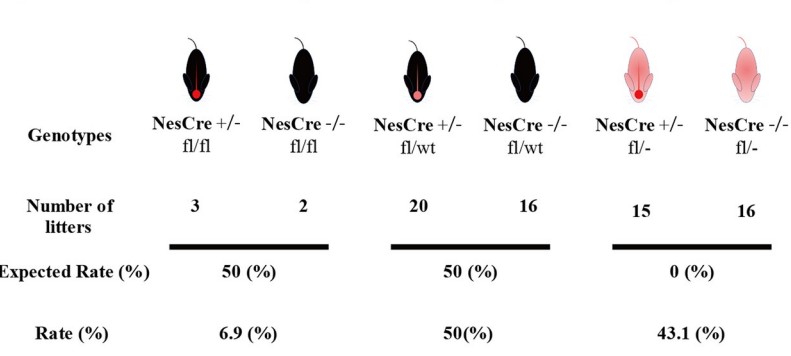

**C.**  **female: NesCre-Glp1r fl/wt X male: Glp1r fl/fl (Total : 15 paris)**

| Genotypes | NesCre +/- fl/fl | NesCre -/- fl/fl | NesCre +/- fl/wt | NesCre -/- fl/wt | NesCre +/- fl/- | NesCre -/- fl/- |
|---|---|---|---|---|---|---|
| Number of litters | 3 | 2 | 20 | 16 | 15 | 16 |
| Expected Rate (%) | 50 (%) | | 50 (%) | | 0 (%) | |
| Rate (%) | 6.9 (%) | | 50(%) | | 43.1 (%) | |

**Rate of germline recombination : 100 %**

**Fig 4. Rate of germline recombination.** Expected and observed progeny from NesCre-Glp1r fl/wt mice crossed with Glp1r fl/fl mice (A); male NesCre-Glp1r fl/wt mice crossed with female Glp1r fl/fl mice (B); female NesCre-Glp1r fl/wt mice crossed with male Glp1r fl/fl mice (C). The number of progenies per genotype, irrespective of Cre status, was indicated. The percentage of germline recombination is delineated from the unexpected observed to expected observed genotypes. Individual data can be found at S1 Data.

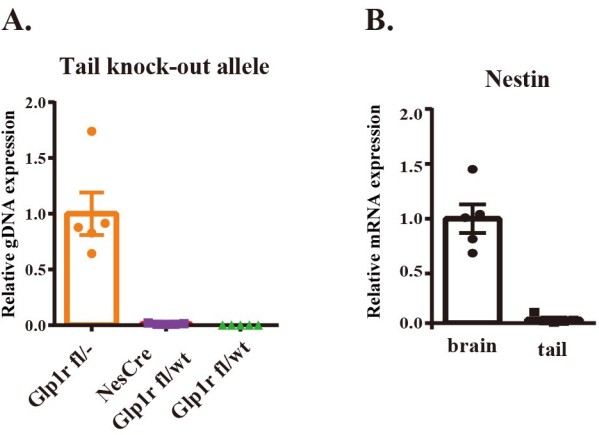

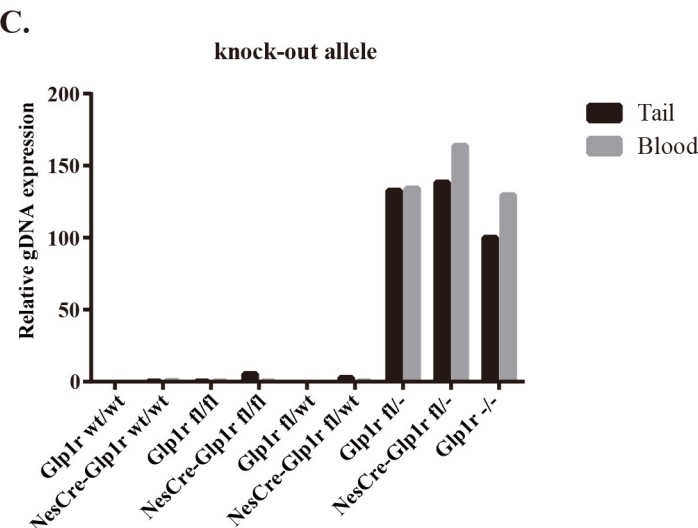

**Fig 5. Somatic Cre activation was observed in the tail of Nestin-Cre with Glp1r floxed mice.** (A) Quantification of tail gDNA level of knock-out allele about NesCre-Glp1r fl/wt mice and Glp1r fl/wt mice against Glp1r fl/- mice; n = 5, mice per group. *p < 0.05 versus Glp1r f/-; #p < 0.05 versus Glp1r fl/wt. (B) Quantitative comparison of *Nestin* mRNA expression in the tail versus the brain of B6 mice; n = 5, mice per group. *p < 0.05 versus brain. (C) Quantification of the expression level of the knock-out allele in tail and blood gDNA for all genotypes used in this experiment.; n = 1, mice per group. Data represent mean ± SEM (A, B). Statistical analysis was Student's t-test. Individual data can be found at S1 Data.

## Investigating the location where knock-out alleles occur in male and female mice

The knock-out alleles observed in our study were likely due to genetic recombination influenced by parental Cre during gametogenesis. However, the specific stage of gametogenesis at which genetic recombination occurs remains unknown. To elucidate this mechanism, we first examined the expression levels of *Nestin* mRNA in various reproductive organs using C57BL/6JJcl mice. In males, the expression level of *Nestin* mRNA relative to the brain was 2.8% in the testes, 24% in the caput epididymis, 5.9% in the corpus epididymis, 52% in the cauda epididymis, 6.3% in sperm collected from the caput epididymis, and 4.2% in sperm collected from the

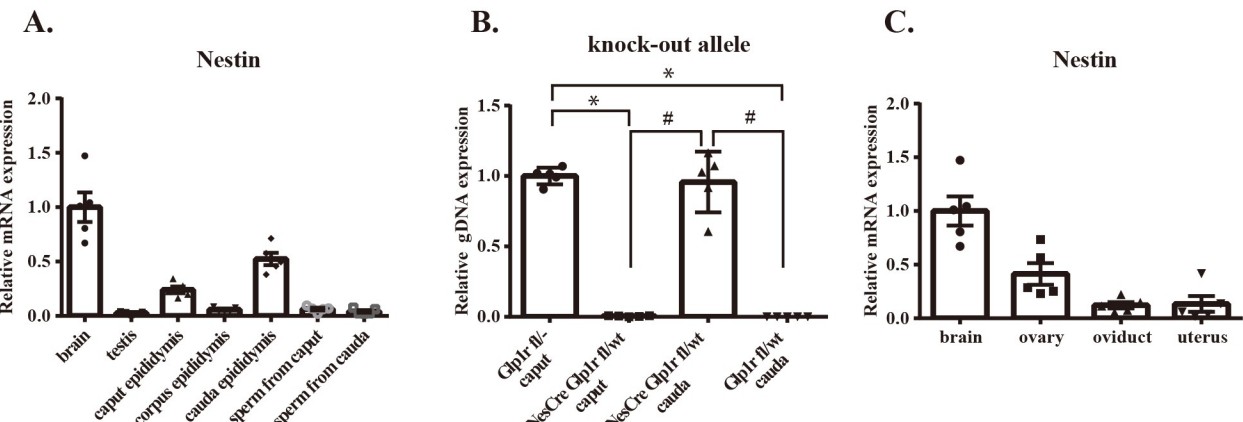

**Fig 6. Comparison of *Nestin* mRNA expression levels in each reproductive organ and confirmation of knockout presence in gametes.** (A) Comparison of *Nestin* mRNA expression levels in the testis, epididymal caput, epididymal corpus, epididymal cauda, sperm collected from the epididymal caput, and sperm collected from the epididymal cauda relative to the brain in C57BL/6JJcl mice. (B) Comparison of knock-out allele expression levels in the gDNA of sperm collected from the epididymal caput of Glp1r fl/- mice, epididymal caput and cauda of NesCre-Glp1r fl/wt mice, and epididymal cauda of Glp1r fl/wt mice.; n = 5, mice per group. *p < 0.05 versus from caput epididymis of Glp1r fl/-; #p < 0.05 versus from cauda epididymis of NesCre-Glp1r fl/wt. Data represent mean ± SEM. Statistical analysis was one-way ANOVA with Tukey's HSD test post-hoc. (C) Comparison of *Nestin* mRNA expression levels in the ovary, oviduct, and uterus relative to the brain in C57BL/6JJcl mice. Individual data can be found at S1 Data.

cauda epididymis (Fig 6A). Next, we collected sperm from the caput and cauda epididymis in NesCre-Glp1r fl/wt mice and compared the expression levels of knock-out alleles in sperm gDNA. In NestinCre-Glp1r fl/wt mice, knock-out alleles were almost undetectable in sperm collected from the caput epididymis. However, the expression levels of knock-out alleles in sperm from the cauda epididymis were comparable to those observed in sperm collected from the caput epididymis of Glp1r fl/- mice. Additionally, we did not observe any knock-out alleles in sperm obtained from the cauda epididymis of Glp1r fl/wt mice (Fig 6B). In light of these results, we compared the gDNA expression levels of the knock-out allele in the spermatogenic cells within the testis, caput epididymal sperm, and cauda epididymal sperm of mice with all genotypes used in this experiment. In mice carrying the knock-out allele such as Glp1r fl/-, NesCre-Glp1r fl/-, and Glp1r -/-, the expression of knock-out allele was observed in testis, caput epididymal sperm, and cauda epididymal sperm to the same extent. Conversely, in mice lacking either Nestin-Cre or the floxed allele, such as Glp1r wt/wt, NesCre-Glp1r wt/wt, Glp1r fl/fl, and Glp1r fl/wt, the expression of knock-out allele was not detected in testis, caput epididymal sperm, and cauda epididymal sperm. However, in NesCre-Glp1r fl/fl mice and NesCre-Glp1r fl/wt mice, the expression of the knockout allele was higher in testis and cauda epididymal sperm than in caput epididymal sperm (S2 Fig). These results suggest that Nestin-cre may be active not only during the spermatogenesis at testis but also during epididymal sperm maturation. On the other hand, in females, the expression level of *Nestin* mRNA relative to the brain was 41% in the ovary, 12% in the oviduct, and 13% in the uterus. However, due to technical limitations, it was challenging to retrieve oocytes and confirm the presence of knock-out alleles (Fig 6C).

## Discussion

In this study, we demonstrated the presence of germline recombination in the Nestin-Cre line by targeting progeny mice that exhibit ectopic recombination but do not inherit Nestin-Cre.

We generated Glp1r -/- mice, which were expected to have homozygous knock-out alleles based on genotyping, and confirmed that this knockout allele is inherited in offspring according to Mendelian genetics. We further confirmed the gene expression and protein expression of Glp1r and demonstrated that the knock-out allele truly reflects a global knockout throughout the body. Based on these findings, it is plausible to propose that the knock-out allele was caused by parental Cre activity at the gametogenesis.

Furthermore, we conducted functional evaluations to assess the effects of the Glp1r depletion in Glp1r -/- mice. There were no significant differences in body weight and food intake on a standard diet between Glp1r -/- and Glp1r fl/fl mice. However, Glp1r -/- mice exhibited impaired glucose tolerance during the ipGTT. These characteristics closely resemble those already reported in the global Glp1r knock-out mice [18–21]. On the other hand, the originally intended neural-specific Glp1r knock-out mice showed no significant differences in body weight and food intake under standard feeding conditions, similar to the global Glp1r knock-out mice model. However, it has been reported that they do not exhibit impaired glucose tolerance during the ipGTT [22, 23]. Failure to accurately evaluate through genotyping can potentially lead to incorrect interpretations of such phenotypic differences.

Germline recombination occurs through genetic recombination mediated by Cre at some stage during gametogenesis. In the case of sperm, they are generated within the testes through spermatogenesis, acquire motility and fertility as they undergo epididymal maturation while passing through the epididymal duct, and are subsequently stored in the epididymal cauda until ejaculation [24]. Genetic recombination mediated by Cre during any of these processes leads to germline recombination. The presence of Nestin in reproductive organs is a matter of debate. While some reports have identified Nestin expression in Leydig cells, seminiferous tubules, spermatocytes, and spermatogonia of the testis, other reports suggest its presence in vascular smooth muscle cells of the epididymis, which are involved in tissue remodeling and repair [14, 17, 25]. There are also various reports regarding the sites of Cre activity in the reproductive organs. Some investigators have observed Cre activity in the testis, whereas others have reported Cre expression in the epididymis [7, 26, 27]. Conversely, James et al. reported a decline in Nestin expression in epididymal cells as individuals grow, with significantly reduced expression observed in adults [25]. In our study, we detected *Nestin* mRNA expression in all male reproductive organs, with notably elevated expression levels in the cauda epididymis. Based on these results, we have noticed the possibility of germline recombination occurring in the epididymis. We examined the occurrence of knock-out alleles at different stages by collecting sperm from the caput and cauda epididymis and analyzing the relative expression levels of knock-out alleles in gDNA. In NesCre-Glp1r fl/wt mice, sperm collected from the cauda epididymis showed an increased expression of the knock-out allele compared to sperm collected from the caput epididymis. This pattern was similarly observed in NesCre-Glp1r fl/fl mice. On the other hand, we also compared the expression levels of the knock-out allele in spermatogenic cells obtained from the testis. While this evaluation comes with the limitation of encompassing various cell types, including germ cells such as spermatocytes and spermatozoa, as well as supporting cells like Sertoli cells, we observed comparable levels of knock-out allele expression in spermatogenic cells as seen in cauda epididymal sperm in both NesCre-Glp1r fl/wt and NesCre-Glp1r fl/fl mice. Consequently, our findings suggest that while the possibility of germline recombination occurring in the testis cannot be ruled out, there is a significant likelihood of Cre-mediated germline recombination during epididymal maturation and storage, especially when Nestin-Cre and the floxed allele are both present. Further investigation is needed to conclusively determine if sperm underwent germline recombination mediated by Cre enzyme expressed in the epididymis. In the case of females, we confirmed the expression of *Nestin* mRNA not only in the ovaries but also in small amounts in

the oviduct and uterus, which is consistent with previous reports [14]. However, it was challenging to determine the specific stage of oocyte development where knock-out occurred due to technical limitations.

Germline recombination has been observed in various studies, but its frequency varies significantly. However, few studies have explored the reasons behind these disparities. In our investigation, we noticed an extremely high occurrence of germline recombination, particularly when the Cre-positive parent was female. Previous research and our experimental findings suggest that several factors contribute to the frequency variations. One such factor is the accessibility of loxP sites for Cre-recombination. A study involving NestinCre-Glp1r floxed mice reported unexpected genotyping events, where loxP sites were introduced on either side of exons 6 and 7 of the *Glp1r* gene. This resulted in nearly a 10% frequency when crossing male Cre-positive and female floxed mice, with an even higher frequency when the Cre-positive parent was female [22]. As both our study and theirs utilized the same Nestin-Cre mice, differences in the location of the loxP sites likely explain the discrepancies. Another factor to consider is the choice of target gene for knockout. It has been observed that germline recombination is more commonly observed when the target gene is related to ectoderm development, which plays a role in gametogenesis and early embryogenesis [7]. Additionally, the level of target gene expression in reproductive organs may influence the sex bias in germline recombination. A previous study has shown that among analyzed Cre driver lines, 82.8% exhibited a sex bias, with 62.1% demonstrating germline recombination through the male parent and 20.1% through the female parent [7]. Similarly, in the Nestin-Cre line, a high frequency of germline recombination was reported, with 86% of progeny from a male Cre-floxed parent and 100% from a female parent showing germline recombination [14]. The frequency of germline recombination may also depend on whether the target gene is involved in embryological development. If global knockout of the gene leads to embryonic lethality or malformed progeny, mice with germline recombination are excluded, potentially underestimating the probability. Conversely, global knockout of the *Glp1r* gene has been shown to have no impact on embryological development and survival [18]. Therefore, the probability of germline recombination calculated in our NestinCre-Glp1r floxed mouse model can be considered an accurate reflection of the true probability.

As mentioned above, in the context of Cre-loxP technology, there is a potential for germline recombination. Therefore, it is essential to identify knock-out alleles through genotyping. However, when the tissues used for genotyping themselves have the possibility of Cre enzyme expression, somatic recombination may also lead to the emergence of knock-out alleles. In such cases, it becomes necessary to establish a genotyping protocol that can distinguish between germline recombination and somatic recombination. In the tail gDNA used for genotyping in this study, qPCR revealed a slight expression of the knock-out allele in both the NesCre-Glp1r fl/fl and NesCre-Glp1r fl/wt genotypes. In contrast, blood gDNA from NesCre-Glp1r fl/fl and NesCre-Glp1r fl/wt mice did not show this expression. This result, likely due to the presence of neurons in the tail, reflects the existence of somatic recombination induced by Nestin-Cre in tail tissues. In previous studies, a method for distinguishing between somatic and germline recombination involved confirming the gene expression levels of the recombination allele through qPCR and establishing cutoff values to differentiate between somatic and germline recombination [28]. When conducting genotyping in tissues where Cre promoter gene expression is expected, it is crucial to carefully configure the protocol of genotyping or consider alternative tissues.

While there is abundant literature on conditional knock-out mice created using Cre-loxP technology, there is a scarcity of information that addresses the awareness and preventive measures of unexpected recombination. The occurrence of unexpected recombination is

influenced by various factors, and simply confirming the expression of Cre promoter in gene databases does not negate the possibility of such recombination events. Therefore, it is crucial to have a comprehensive understanding of these potential risks and corresponding solutions when utilizing this technology in experimental settings.

## Conclusions

We have found that Nestin-Cre mice crossed with Glp1r floxed mice exhibit unexpected germline recombination with remarkable efficiency. Undetected germline recombination may critically lead to misinterpretations of results from conditional knock-out mice, particularly when it is challenging to discern phenotypes in heterozygous or homozygous knock-out mice. To accurately interpret experimental outcomes, it is essential to devise appropriate breeding strategies and precise genotyping methods.

## Methods

### Mice

All animal experimentation was conducted in accordance with Kyushu University Animal Experiment Regulations (No. A23-086, No. A21-108). Mice were housed in a 12-hour light, 12-hour dark cycle at approximately 22˚C with a standard diet (CRF-1; Oriental Yeast, Japan) and water ad libitum. C57BL/6JJcl mice were purchased from CLEA Japan, Inc. (Japan). Glp1r floxed mice were purchased from TRANS GENIC INC. (Japan). These mice were generated by introducing a loxP site on either side of exons 4 and 5 of the *Glp1r* gene (S3 Fig). We generated conditional knock-out of the Glp1r by breeding the floxed mice with Nestin-Cre mice (B6. Cg-Tg [Nes-cre]1Kln/J line, stock number 003771, Jackson Laboratory). Genotyping was conducted on tail biopsy collected at 4-week-of-age. Mice (age range 12–20 weeks) for tissue collection were litter-matched and group-housed. Mice were anesthetized by isoflurane before decapitation. Tissues were collected and frozen for later analysis.

### Genotyping

The extraction of gDNA from tail samples was performed using the Hot Shot method and the extraction of gDNA from blood samples was performed using High Pure PCR Template Preparation Kit (Roche, Switzerland). Polymerase chain reaction (PCR) was performed using KOD One™ PCR Master Mix -Blue- (TOYOBO, Japan). Primer sequences and PCR conditions are detailed in the S2 Table.

### Quantitative real-time PCR

To quantify Glp1r mRNA expression, total RNA of the tissues was extracted using ISOGEN (NIPPON GENE Co., Japan) according to the manufacturer's protocol. RNA was reverse transcribed with PrimeScript™ RT reagent kit (TAKARA BIO Inc., Japan). To assess the relative quantity of knock-out alleles, gDNA was extracted from tails and sperms using DNeasy Blood & Tissue Kit (QIAGEN Inc., Netherlands). Quantitative RT-PCR was performed using StepOnePlus Real-time PCR System with Go Taq® Green Master Mix (Promega, USA). mRNA expression levels and gDNA levels were normalized to those of 36b4. Relative expression level of the Glp1r gene was determined using the $2^{-\Delta\Delta Ct}$ method.

Primer sequences: Glp1r (forward 5′– CAT GTG TAC CGG TTC TGC AC –3′, reverse 5′– CAA GGC GGA GAA AGA AAG TG –3′). Nestin (forward 5′– GCA GGA GAA GCA GGG TCT AC –3′, reverse 5′– GGG GTC AGG AAA GCC AA –3′). 36b4 (forward 5′– GGC CCT GCA CTC TCG CTT TC –3′, reverse 5′– TGC CAG GAC GCG CTT GT –3′).

Glp1r knock-out (forward: 5′– ACA CAC ACA CAC TAT AAC AGT GGA TGG –3′, reverse: 5′– CAC AGT CTC ATG GCC AGG AG –3′).

## Western blotting analysis

The total protein of 30 μg was fractionated by sodium dodecyl sulfate-polyacrylamide gel electrophoresis, and then transferred to polyvinylidene fluoride membranes. Primary antibodies used in Western blotting were as follows: anti-GLP-1R (1:1000, ab218532, Abcam, UK), Actin antibody(I-19): sc-1616 HRP (B1815, Santa Cruz, USA). Secondary antibodies were horseradish peroxidase-conjugated goat anti-rabbit lgG (1:20000 Santa Cruz, USA). The results were visualized using an enhanced chemiluminescence system (ECL™ prime, Amersham, UK). Densitometric analysis was conducted using ImageJ software (NIH, USA).

## Metabolic and biochemical studies

For metabolic and biochemical studies, we used only male mice. Body weights were monitored once a week from 10 weeks to 20 weeks. Food intake was individually measured for 3 consecutive days at 11 weeks using the mouse feeder MF-4S (Shin Factory, Japan). Blood glucose and serum insulin levels were measured with Stat Strip XP3 (Nipro, Japan) and ELISA (Morinaga Institute of Biological Science, Japan), respectively. For intraperitoneal glucose tolerance test (ipGTT), mice were fasted for 16 h with free access to water followed by intraperitoneal glucose injection (1.5 g/kg). We measured blood glucose at 0, 15, 30, 60, and 120 min after glucose injection, and serum insulin at 0, 15, and 30 min after glucose injection. To investigate the effects of Glp1r signaling on anorectic responses, mice received liraglutide (Novo Nordisk; 1 mg/kg) or saline (10 ml/kg) 1 hour prior to the dark cycle, and food consumption was measured after 24 hours using the mouse feeder MF-4S.

## Statistical analysis

Data are expressed as mean ± standard error of the mean (SEM). The statistical significance was analyzed by Student's t-test, or two-way repeated measures ANOVA with Bonferroni corrected Student's t-test, or one-way ANOVA followed by Tukey's honestly significant difference (HSD) test for multiple comparisons. P-value <0.05 was considered statistically significant. Data were analyzed by using JMP Pro 16.0 software (SAS Institute Japan Inc., Japan).

## Supporting information

**S1 Fig. Genotyping results using blood samples for Glp1r wt allele, floxed allele, knock-out allele, and Nestin-Cre allele.** Genotyping was performed on blood samples collected from mice of each identified genotype obtained through tail genotypingd using NestinCre primer, primer set 1, and primer set 2. Underlying raw images can be found at S1 Raw images. (TIF)

**S2 Fig. Spermatogenic cell site-specific comparison of knock-out allele expression levels.** Comparative analysis of gDNA knock-out allele expression levels in spermatogenic cells within the testis, caput epididymal sperm, and cauda epididymal sperm across Glp1r wt/wt, NesCre-Glp1r wt/wt, Glp1r fl/fl, NesCre-Glp1r fl/fl, Glp1r fl/wt, NesCre-Glp1r fl/wt, Glp1r fl/-, NesCre-Glp1r fl/-, and Glp1r -/- mice. The expression level of each knock-out allele is represented as a numerical value relative to a reference value of 1, which corresponds to the expression level in cauda epididymal sperm of Glp1r -/- mice. n = 1, mice per group. Individual data can be found at S1 Data. (TIF)

**S3 Fig. Construction of Glp1r conditional targeting vector.** Glp1r genomic locus and targeting design. The targeting constructs incorporated loxP sites flanking exons 4 and 5 and a phosphoglycerate kinase promotor-neomycin (PGKp-Neo)-resistance cassette flanked by FRT sites.
(TIF)

**S1 Table. Average period from mating to birth, the number of litters, and the number of deaths till weaning.** This table shows the average period from mating to birth, the number of litters, and the number of deaths till weaning in several mating patterns (Glp1r fl/fl × Glp1r fl/fl, Glp1r fl/- × Glp1r fl/-, Glp1r -/- × Glp1r -/-, female NesCre-Glp1r fl/wt × male Glp1r fl/fl, female Glp1r fl/fl × male NesCre-Glp1r fl/wt,). Individual data can be found at S1 Data.
(TIF)

**S2 Table. Primer sequences and PCR conditions for genotyping.** This table shows the primer sequence and PCR conditions for primer set1, primer set2 and Nestin-Cre primer.
(TIF)

**S1 Data. Data underlying Figs 2–6, S2 Fig, and S1 Table.**
(XLSX)

**S1 Raw images. Original gel and images related to Figs 1B and 2C, and S1 Fig.**
(ZIP)

## Acknowledgments

We thank the members of Ogawa Laboratory for the helpful discussions. The authors also thank Ms. Makiko Kamihashi and Ms. Aya Fujimoto (Department of Medicine and Bioregulatory Science, Graduate School of Medical Sciences, Kyushu University, Fukuoka, Japan), for technical assistance. We also appreciate the technical assistance from The Research Support Center, Kyushu University Graduate School of Medical Sciences.

## Author Contributions

**Conceptualization:** Yusuke Kajitani.

**Data curation:** Takashi Miyazawa.

**Formal analysis:** Yusuke Kajitani, Nao Kajitani.

**Funding acquisition:** Takashi Miyazawa.

**Investigation:** Yusuke Kajitani, Takashi Miyazawa, Tomoaki Inoue.

**Project administration:** Yusuke Kajitani.

**Resources:** Takashi Miyazawa.

**Supervision:** Takashi Miyazawa.

**Validation:** Yusuke Kajitani, Takashi Miyazawa.

**Visualization:** Yusuke Kajitani, Nao Kajitani.

**Writing – original draft:** Yusuke Kajitani, Takashi Miyazawa, Nao Kajitani, Masamichi Fujita, Yukina Takeichi, Yasutaka Miyachi, Ryuichi Sakamoto.

**Writing – review & editing:** Yusuke Kajitani, Takashi Miyazawa, Yoshihiro Ogawa.

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
