## [Decision Letter · Decision Letter 0]

29 Aug 2023

PONE-D-23-21443High frequency of germline recombination in Nestin-Cre transgenic mice crossed with Glucagon-like peptide 1 receptor floxed micePLOS ONE

Dear Dr. Miyazawa,

Thank you for submitting your manuscript to PLOS ONE. After careful consideration, we feel that it has merit but does not fully meet PLOS ONE’s publication criteria as it currently stands. Therefore, we invite you to submit a revised version of the manuscript that addresses the points raised during the review process.

While your study is interesting and adds value to the domain knowledge, there are some technical issues that need to be addressed. Genotyping of the mice should be provided. Quantification of protein expression in the Western blots be performed. Detailed information on the genotyping strategy is required.  Additional experiments focusing on the caput sperm and the occurence of recombination in these germ cells to be proven.

We look forward to receiving your revised manuscript.

Kind regards,

Suresh Yenugu

Academic Editor

PLOS ONE

Reviewers' comments:

Reviewer's Responses to Questions

**Comments to the Author**

1. Is the manuscript technically sound, and do the data support the conclusions?

Reviewer #1: Partly

Reviewer #2: Partly

2. Has the statistical analysis been performed appropriately and rigorously? 

Reviewer #1: Yes

Reviewer #2: Yes

3. Have the authors made all data underlying the findings in their manuscript fully available?

Reviewer #1: Yes

Reviewer #2: Yes

4. Is the manuscript presented in an intelligible fashion and written in standard English?

Reviewer #1: Yes

Reviewer #2: Yes

5. Review Comments to the Author

Reviewer #1: This study shows the results of an attempt to generate neuron-specific Glp1r KO mice using Nestin-Cre driver mice. However, the authors discovered that the null allele was present with or without Nestin-cre expression in tail DNA, suggesting the occurrence of germline recombination. The authors found that germline recombination occurs during the process of epididymal maturation in male mice.

Major points:

Authors should show the Cre-genotyping results, and indicate Cre+ or - genotype, for the samples genotyped for Glp1r in Fig. 1D.

The authors should label lane 1 and 2 in Fig. 1D.

Is the nestin promoter expressed in various tissues tested in Fig. 2B?

The western blots in Fig. 2C should be quantified as a Glp1r band is observable in Glp1r-/- lanes. A darker, non-adjusted exposure should be shown for the stomach blots. Molecular weight markers are missing.

Line 146-148; from this statement, it is unclear whether fasting blood glucose or blood glucose levels are shown in Fig. 3D.

How do the results shown in Fig. 3F of the HOMA-IR and HOMA-b tests compare to full-body Glp1r KO animals?

Examples of genotyping results (as done in Fig. 1) would be complementary to diagrams shown in Fig. 4.

Could the authors indicate the genotyping strategy employed in Fig. 4 to detect the floxed allele (ie primer set 1 vs 2). The authors previously indicated difficulty in detection of the floxed vs WT allele using primer set 2 (Fig.1). This technical issue may prevent accurate results in determining the frequency of recombination, depending on the genotyping strategy employed in Fig. 4. Was Cre genotype established?

Fig. 5C; the results in Fig. 1 indicate that knockout alleles were genotyped for to determine the identification of fl/- mice. Please clarify further.

Reviewer #2: The manuscript written by Kajtani et. al. describes the generation of mice with KO of the Glp1r gene irrespective of the presence of Cre recombinase. They also found that the KO trait was a global event. Initially, they breed the mice with nestin Cre background that should generate a neuron specific KO trait. But they found the generation of a nonspecific global KO trait of the Glp1r gene. Their further investigation found that the KO event was a result of germ-line recombination.

The manuscript is well written without the use of unnecessary jargon.

The manuscript points out important phenomena of which the researchers of this field should be well aware.

1. The authors should have performed genotyping from gDNA isolated from blood rather than tail samples as there are chances of getting false positives during detection due to possible expression of nestin-cre in the tail region. The blood data will represent a more global perspective in this regard.

2. The caudal part of the epididymis serves as a reservoir of total sperm pool where all the knockout allele carrying spermatozoa will be present in abundance and will be comparable with the number of spermatozoa present in the caput region. So, the statement mentioned in lines 211-215 needs to be further strengthened by other experiments. Rather the author should focus more on the spermatogenic cells where most of the recombination steps are known to occur.

3. The author should explain the mating schematics and results and methods in a more understandable manner.

4. In the discussion in the lines 263-267 the authors are trying to state that in nestin Cre-Glp1rfloxed mice they observed germline recombination where as they have to be clear whether the event was observed in nestin Cre-Glp1r fl/fl or nestin Cre-Glp1r fl/f- mice as in case of nestin Cre-Glp1r fl/f- mice due to genetic recombination and meiosis it is possible to get Glp1r knockout allele. They also have to experimentally validate this phenomenon using nestin Cre-Glp1r fl/fl mice also.

5. To prove the claim that the recombnatio event occur in the epididymis needs further investigation.

6. PLOS authors have the option to publish the peer review history of their article (what does this mean?). If published, this will include your full peer review and any attached files.

Reviewer #1: No

Reviewer #2: No

---

## [Author Response · Author response to Decision Letter 0]

16 Nov 2023

Thank you very much for your valuable comments and suggestions. We have explained our standpoint together with point-by-point responses to the questions raised, as indicated in separate sheets. We believe that we have satisfactorily responded to all of the critiques raised by the Reviewers.

---

## [Decision Letter · Decision Letter 1]

4 Dec 2023

High frequency of germline recombination in Nestin-Cre transgenic mice crossed with Glucagon-like peptide 1 receptor floxed mice

PONE-D-23-21443R1

Dear Dr. Miyazawa,

We’re pleased to inform you that your manuscript has been judged scientifically suitable for publication and will be formally accepted for publication once it meets all outstanding technical requirements.

Kind regards,

Suresh Yenugu

Academic Editor

PLOS ONE

Additional Editor Comments (optional):

Reviewers' comments:

Reviewer's Responses to Questions

**Comments to the Author**

1. If the authors have adequately addressed your comments raised in a previous round of review and you feel that this manuscript is now acceptable for publication, you may indicate that here to bypass the “Comments to the Author” section, enter your conflict of interest statement in the “Confidential to Editor” section, and submit your "Accept" recommendation.

Reviewer #1: All comments have been addressed

Reviewer #2: All comments have been addressed

2. Is the manuscript technically sound, and do the data support the conclusions?

Reviewer #1: Yes

Reviewer #2: Yes

3. Has the statistical analysis been performed appropriately and rigorously? 

Reviewer #1: Yes

Reviewer #2: Yes

4. Have the authors made all data underlying the findings in their manuscript fully available?

Reviewer #1: Yes

Reviewer #2: Yes

5. Is the manuscript presented in an intelligible fashion and written in standard English?

Reviewer #1: Yes

Reviewer #2: Yes

6. Review Comments to the Author

Reviewer #1: (No Response)

Reviewer #2: (No Response)

7. PLOS authors have the option to publish the peer review history of their article (what does this mean?). If published, this will include your full peer review and any attached files.

Reviewer #1: No

Reviewer #2: No

---

## [Editor Report · Acceptance letter]

11 Dec 2023

PONE-D-23-21443R1 

High frequency of germline recombination in Nestin-Cre transgenic mice crossed with Glucagon-like peptide 1 receptor floxed mice 

Dear Dr. Miyazawa:

I'm pleased to inform you that your manuscript has been deemed suitable for publication in PLOS ONE. Congratulations! Your manuscript is now with our production department. 

Kind regards, 

on behalf of

Dr. Suresh Yenugu 

Academic Editor

PLOS ONE